# Construction and Simulation Analysis of Epidemic Propagation Model Based on COVID-19 Characteristics

**DOI:** 10.3390/ijerph20010132

**Published:** 2022-12-22

**Authors:** Sheng Bin

**Affiliations:** College of Computer Science & Technology, Qingdao University, Qingdao 266071, China; binsheng@qdu.edu.cn

**Keywords:** public health, COVID-19, epidemic propagation model, data analysis, model simulation

## Abstract

This paper proposes the epidemic propagation model SEAIHR to elucidate the propagation mechanism of the Corona Virus Disease of 2019 (COVID-19). Based on the analysis of the propagation characteristics of COVID-19, the hospitalization isolation state and recessive healing state are introduced. The home morbidity state is introduced to consider the self-healing of asymptomatic infected populations, the early isolation of close contractors, and the impact of epidemic prevention and control measures. In this paper, by using the real epidemic data combined with the changes in parameters in different epidemic stages, multiple model simulation comparative tests were conducted. The experimental results showed that the fitting and prediction accuracy of the SEAIHR model was significantly better than the classical epidemic propagation model, and the fitting error was 34.4–72.8% lower than that of the classical model in the early and middle stages of the epidemic.

## 1. Introduction

With the rapid spread of COVID-19 worldwide, there have been more than 200 countries and regions with COVID-19 cases. As of 3 November 2022, there are 608,328,548 confirmed cases of COVID-19 and 6,501,469 deaths in the world, and the data is still increasing. Relevant studies [1,2,3] have shown that novel COVID-19 not only has a long incubation period but also has infectivity in the incubation period, and there are recessive latencies in the transmission process, that is to say, asymptomatic and infectious groups after the incubation period. Therefore, it is necessary to establish a propagation model that can accurately describe the transmission mechanism of COVID-19, combined with the transmission characteristics of COVID-19 and the normalized epidemic prevention and control measures.

Since the outbreak of COVID-19, researchers have conducted a lot of relevant research on epidemic propagation models. Liu et al. [4] constructed the SIRU model, which first proposed the number of unreported symptomatic cases, and predicted the trend of epidemic transmission in different levels of prevention and control measures. Li et al. [5] estimated the distribution of key epidemiological delay time, the doubling time of infectious diseases, and the basic reproduction number based on the data of the earliest 425 confirmed cases. Siqueira et al. [6] established a dynamic propagation model, including infectivity in the latent period, and estimated the basic regeneration number by using a simulation. Sora et al. [7] considered the influence of COVID-19 latency based on the SEIR model and predicted the inflection point by using simulation experiments. Macalisang et al. [8] considered the symptomatic and asymptomatic states and the decline of immunity and established a propagation model with seven compartments. Based on the SEIR model, Nasir [9] considered the containment effect of isolation measures on the epidemic situation, solved the model using the Euler numerical method, and evaluated and predicted the epidemic situation. Efimov et al. [10] separated the cured and the dead on the basis of the traditional SEIR model and analyzed the epidemic situation development trend in eight different regions, considering the changes in and uncertainties of model parameters. Korolev [11] constructed a spatio-temporal coupling model of infectious diseases based on SEIRD-GEOCA and incorporated spreading and random migration diffusion strategies into the model to simulate the spatio-temporal diffusion of the COVID-19 epidemic. Anna et al. [12] used the SEIR model with vaccination and isolation factors as model parameters and used the generation matrix method to obtain the basic reproduction number and global stability of the COVID-19 distribution model. Ngonghala et al. [13] used the improved SEIR model and the prediction of key driving covariates (pneumonia seasonality, mobility, detection rate, and per capita mask usage rate) to assess the social distance requirements and mask use level. Li et al. [14] considered the data lag and established a propagation model based on time-varying parameters to predict the development of the COVID-19 epidemic. Asamoah et al. [15] proposed a new epidemic model, which considers the economic evaluation of control terms in an enclosed population. De et al. [16] proposed a newly proposed SEIADR model which incorporates asymptomatic and dead-infective subpopulations into the standard SEIR model. Esquível et al. [17] proposed an improved SIR model with a compartment for deaths and with regime switching to model pandemic mitigation measures.

Existing research shows that COVID-19 shows more special propagation characteristics than previous infectious diseases, resulting in faster propagation speed, wider propagation range, and higher propagation risk. In the context of such a severe global epidemic situation, it is urgent to study the propagation dynamics model based on the characteristics of COVID-19 and prevention and control measures in combination with the special propagation characteristics of COVID-19, so as to provide theoretical support for epidemic prevention and control. However, the parameters of the existing models are mostly fixed, and the applicability of the existing models is gradually weakened with the variation of viruses and the change in epidemic prevention and control measures. Therefore, this paper increases the robustness of the model by controlling parameter changes, refining the model according to the actual prevention and control measures, and improving the model to analyze the evolution process of COVID-19 more accurately.

The main contributions of the proposed model in this paper include the following:(1)In consideration of the existence of recessive latent populations and their COVID-19 infectivity, the recessive healing compartment is established to facilitate the simulation and analysis of the impact of such populations on the process of epidemic development.(2)In consideration of the early isolation of the latent population for epidemic prevention and control, the hospitalization isolation compartment is established.(3)Considering the time-varying characteristics of model parameters, the cure rate and mortality rate are set as daily change parameters, and the infection rate and other parameters are deemed to be phased changes.

## 2. A New Epidemic Propagation Model Based on COVID-19 Characteristics

The traditional SEIR model divides the population into four categories: susceptible, exposed, infected, and removed. Susceptible (*S*) represents the uninfected healthy population; they can be infected if they come into contact with an infected person. Exposed (*E*) refers to the people who have no symptoms but are infected with the virus. The virus has not yet occurred in such people and is not infectious. Infected (*I*) refers to people who have symptoms and are infectious. Such people can infect susceptible people through physical contact. Removed (*R*) refers to the people who have been cured or died because of their infection. The cured people have antibodies and will not be infected again. The transition between the states of the SEIR model is shown in Figure 1.

The propagation dynamics equation of the SEIR model is shown in Equation (1)
(1)$$\left\{\begin{array}{l} \frac{dR_{t}}{dt}=-\lambda \varepsilon S_{t},\\ \frac{dE_{t}}{dt}=\lambda \varepsilon S_{t}-\rho E_{t},\\ \frac{dI_{t}}{dt}=\rho S_{t}-\alpha I_{t},\\ \frac{dR_{t}}{dt}=\alpha I_{t},\\ \frac{dS_{t}}{dt}=\lambda \varepsilon \rho \alpha S_{t}. \end{array}\right.$$$S_{t}$, $E_{t}$, $I_{t}$, and $R_{t}$ represent the number of susceptible, exposed, infected, and removed persons at time t, respectively. λ represents the average number of persons that come into contact with infected people at each moment, ε represents the proportion of susceptible persons in the total population, ρ represents the proportion of the population becoming infected at each moment, and α represents the probability of infected people transforming into removed people.

With the development of COVID-19 and the deepening of related research, the spread of COVID-19 has shown the following characteristics: (1) COVID-19 virus has an incubation period, in which there are no obvious symptoms, and it is infectious. (2) There are a certain number of self-healing persons among the exposed population, who are asymptomatic but infectious after the incubation period, and will recover after the onset cycle. (3) With the progress of epidemic prevention and control, the exposed persons can be detected and hospitalized for isolation treatment before the disease’s onset. According to the above description, it can be found that the traditional SEIR model is not completely consistent with the actual propagation characteristics of COVID-19. Therefore, based on the traditional SEIR model, through introducing three new states, this paper constructs a SEAIHR model that is more consistent with the actual propagation characteristics of COVID-19.

According to the actual spread of COVID-19, the SEAIHR model is based on the following assumptions: (1) Population mobility is not considered during the epidemic. (2) All hospitalized infected persons are isolated and not infectious. (3) The affected people will protect themselves before admission, and their infectivity could be ignored. (4) The natural death and birth rates of the population during the pandemic are ignored.

In the SEAIHR model, there are eight compartments: susceptible, effective exposed, arcane exposed, illness, hospitalized, cured, removed, and dead. Susceptible (*S*) refers to a healthy population that is easily infected. Effective exposure (*E*) refers to a population with symptoms and infectivity after the incubation period. Arcane exposed (*A*) refers to the population that has no symptoms and is still infectious after the incubation period. Illness (*I*) refers to the population that is not hospitalized temporarily after the incubation period. Hospitalized (*H*) refers to the population that is isolated for treatment and is not infectious. Cured (*C*) refers to the recessive latent population that recovers after the onset cycle. This group does not show obvious symptoms and is not included in the officially announced cure cases. Removed (*R*) refers to the population that has been cured after hospitalization and isolation. Dead (*D*) refers to the population that died after hospitalization and isolation. The transition rules between states are as follows: (1) The susceptible population (*S*) is infected by being exposed to individuals with a certain probability of infection and is transformed into effective exposed (*E*) and arcane exposed (*A*) populations. (2) In the initial stage of the epidemic, exposed persons were not tracked and isolated in time, and arcane exposed people (*A*) were transformed into cured (*C*) people. While Effective exposed (*E*) people were hospitalized for isolation after the onset of their COVID-19 symptoms, they became hospitalized (*H*) patients. (3) In the middle and late stages of the epidemic, effective exposed (*E*) and arcane exposed people (*A*) were tracked and isolated for hospitalization and became hospitalized patients (*H*). Through isolation treatment, they became removed (*R*) or dead (*D*) people. When the population was in a state of cure or death, their state would not change. The transition between the states of the SEAIHR model is shown in Figure 2.

Based on the above transition rules between states and system dynamics, the propagation dynamic equation of the SEAIHR model is shown in Equation (2):(2)$$\;\left\{\;\begin{array}{l} \frac{dS_{t}}{dt}=-\left(E_{t}\lambda_{1}+A_{t}\lambda_{2}\right),\\ \frac{dE_{t}}{dt}=\left(1-m\right)\left(E_{t}\lambda_{1}+A_{t}\lambda_{2}\right)-E_{t}\beta_{1}-E_{t}\mu ,\\ \frac{dA_{t}}{dt}=m\left(E_{t}\lambda_{1}+A_{t}\lambda_{2}\right)-A_{t}\beta_{3}-A_{t}\beta_{2},\\ \frac{dI_{t}}{dt}=E_{t}\mu -I_{t}\rho ,\\ \frac{dH_{t}}{dt}=E_{t}\beta_{1}+I_{t}\rho -H_{t}\left(P_{r}+P_{d}\right),\\ \frac{dC_{t}}{dt}=A_{t}\beta_{3},\\ \frac{dR_{t}}{dt}=H_{t}P_{r},\\ \frac{dD_{t}}{dt}=H_{t}P_{d}. \end{array}\;\right.$$$S_{t}$, $E_{t}$, $A_{t}$, $I_{t}$, $H_{t}$, $C_{t}$, $R_{t}$, and $D_{t}$ represent the number of people in each state at time t, respectively.

The dynamic model includes ten parameters, some of which can be determined by estimation or official data, and these parameters are direct parameters. Other parameters cannot be obtained or estimated directly. These parameters are indirect parameters, and the indirect parameters are fitted to their optimal values by parameter inversion. The parameters are explained in Table 1.

Because the arcane exposed population and effective exposed population can be distinguished only after being confirmed, $\beta_{1}$ and $\beta_{2}$ can only be set to the same value during the simulation.

When the value of {$\beta_{1}$, $\beta_{1}$, $\beta_{1}$, μ, m} is known, the optimal value of {$\lambda_{1}$, $\lambda_{2}$, ρ} can be obtained by solving Equation (3), and the optimization objective is the deviation between the actual data and the simulated data.
(3)$$\underset{\lambda_{1},\lambda_{2},p}{\min}\;\|H\left(\lambda_{1},\lambda_{2},\rho ,t\right)-H_{data}\;{\left(t\right)\|}_{2}+\|R\left(\lambda_{1},\lambda_{2},\rho ,t\right)-R_{data}{\left(t\right)\|}_{2}+\|D\left(\lambda_{1},\lambda_{2},\rho ,t\right)-D_{data}{\left(t\right)\|}_{2}$$

The recovery rate and death rate are regarded as daily variation parameters. $P_{r}$ and $P_{d}$ are the proportion of the number of newly recovered people and the new death toll in the previous day’s hospitalization and isolation. In this paper, the nonlinear least squares method is used for fitting, that is, the parameters are solved by solving the local minimum value of Equation (4), and the fitting equation uses a power function.
(4)$$F\left(x\right)=\frac{1}{2}\sum {\left(y_{1}\left(x\right)-y_{2}\right)}^{2}$$
where $y_{1}\left(x\right)$ represents the objective function value to fit, $y_{2}$ represents the actual value.

Through finding the optimal independent variable x, the sum of the residual squares of each point $\left(y_{1}\left(x\right)-y_{2}\right)$ is minimized to obtain the fitted curve. This paper uses the actual data of Wuhan, China, to remove the time points with large fluctuations on 12–13 February 2020. The fitting results are shown in Table 2.

The recovery rate and death rate in Wuhan show a trend of change over time. The two parameters are set as time-varying parameters to simulate the real situation of the epidemic development more accurately. The fitting results are shown in Figure 3 and Figure 4.

## 3. Simulation Results and Analysis

This paper verifies the accuracy of the SEAIHR model by comparing the simulation results with the real epidemic data. The experiment used the root-mean-square error (RMSE) and the average relative error (MAPE) of the fitting data and the number of confirmed cases as the model accuracy indicators, which are defined as follows:(5)$$\mathrm{RMSE}=\sqrt{\frac{1}{t}\sum_{i=1}^{t}{\left(H\left(i\right)-R\left(i\right)\right)}^{2}{}^{}}.$$
(6)$$\mathrm{MAPE}=\frac{100\%}{t}\sum_{i=1}^{t}\left|\frac{R\left(i\right)-H\left(i\right)}{R\left(i\right)}\right|.$$
where $R\left(i\right)$ represents the real data of the number of confirmed cases, $H\left(i\right)$ represents the simulation data of the number of confirmed cases.

The smaller the values of RMSE and MAPE, the closer the fitting results are to the real data, the higher the accuracy of the model fitting and prediction, and the better the simulation effect.

### 3.1. COVID-19 Retrospective Analysis

The experiment was, according to the COVID-19 epidemic data of Wuhan, conducted in China from 22 January 2020 to 26 April 2020 (the data were released by Wuhan Municipal Health Commission) (http://wjw.wuhan.gov.cn (accessed on 26 April 2020)). Based on the SIR model, SEIR model, and SEAIHR model, a simulation comparison experiment is carried out to verify the correctness of the SEAIHR model. On 12 February 2020, Wuhan reported suspected cases for the first time, which means that most of the arcane exposed persons could be tracked by Wuhan’s medical institutions at this time, and the relevant reports of asymptomatic infections begin to appear in various regions. Since 26 February 2020, the number of newly diagnosed cases has decreased significantly, and the number of cured cases has increased significantly. Until 27 April 2020, the number of confirmed cases was cleared. Therefore, the experiment would be divided into three stages:

Stage 1 (From 22 January to 11 February 2020): During the epidemic outbreak period, the number of new infections increased daily, the medical institutions were limited, and the prevention and control measures were not mature, which led to the rapid spread of the disease.

Stage 2 (From 12 February to 26 February 2020): The epidemic situation was under effective control. At this time, suspected cases had been reported, and the daily increase rate had slowed down.

Stage 3 (From 26 February to 27 April 2020): During the recession period of the epidemic, the number of existing confirmed cases had reached a peak and showed a downward trend. The cure rate was greater than the incidence rate, and the epidemic showed a declining trend.

#### 3.1.1. Epidemic Fitting Results Analysis of Stage 1

The official data released at the initial stage of the COVID-19 epidemic did not include the number of exposed persons and the number of patients. According to the incubation period interval and the time from onset to hospitalization pointed out in the relevant literature [18,19], this paper iteratively deduces the initial values of exposed persons and patients, in which the proportion of arcane exposed persons *m* is set according to their average proportion in the total number of exposed persons.

In order to obtain estimates of the standard errors of parameters in the SEAIHR model, 500 simulations of the SEAIHR model are obtained. The standard error can then be estimated from these simulations by using the root-mean-square of the errors in the parameter estimates.

The fitting results for Wuhan from 22 January to 11 February 2020 are shown in Figure 5.

As shown in Figure 5, the three models showed an accelerated upward trend at the initial stage of the epidemic, but the upward trend of the SIR model and SEIR model far exceeded the development of the real epidemic, and the overall fitting trend of the SEAIHR model was the best.

The evaluation results of the three models in stage 1 are shown in Table 3.

According to the evaluation results in Table 3, the fitting accuracy of the SEAIHR model has been significantly improved. The root-mean-square error (RMSE) is 59% smaller than the SIR model and 53% smaller than the SEIR model. The average relative error (MAPE) is 37.3% smaller than the SIR model and 34.4% smaller than the SEIR model.

When the best fitting result of stage 1 is obtained, the parameter values of the SEAIHR model are set as shown in Table 4.

Among the parameters in stage 1, the infection rates of the two exposed types were 0.491 and 0.433, respectively, and the $\lambda_{1}$ value is slightly higher than $\lambda_{2}$. It shows that due to the limited understanding and medical ability of the COVID-19 virus in the early stage of the epidemic and the huge pressure of admission, the exposed persons spread freely among the population, and the viral load of the arcane exposed persons is lower than that of the effective exposed ones. At this time, the epidemic is in a rapid outbreak period. In addition, due to the serious lack of detection means and tracking ability, the self-healing of arcane exposed persons was not included in the official data, and the effective exposed persons were hospitalized after the onset of the disease. However, due to the limited medical capacity in the early epidemic period, the incidence of hospitalization was only 0.09, indicating that there was a long time delay from the onset to hospitalization, which further led to a low cure rate and a high death rate.

#### 3.1.2. Epidemic Fitting Results Analysis of Stage 2

Since 12 February 2020, Wuhan has started to count suspected cases, which shows that most of the potential cases can be traced. At the same time, the number of designated hospitals has increased from 2 to 48, and 16 new shelter hospitals have been built, which means that the epidemic trend would have an inflection point. The simulation results of stage 2 are shown in Figure 6 and Figure 7.

It can be seen from Figure 6 that the SEAIHR model accurately reproduced the inflection point and decline trend of the COVID-19 epidemic in Wuhan from 12 February 2020 to 26 February 2020.

The evaluation results of the three models in stage 2 are shown in Table 5.

Table 5 shows that RMASE and MAPE values of the SEAIHR model are reduced by 52.6% and 52.8%, respectively, compared with the SIR model and by 73.6% and 76.8%, respectively, compared with the SEIR model, which fully reflects the rationality and accuracy of the SEAIHR model.

As shown in Figure 7, the number of effective exposed persons decreased by 70%, and the number of arcane exposed persons decreased to 143, indicating that the epidemic spread was effectively controlled with the increase of tracking and isolation intensity and the enhancement of self-protection awareness.

When the best fitting result of stage 2 is obtained, the parameter values of the SEAIHR model are set, as shown in Table 6.

The parameters in Table 6 show that the infection rates of effective exposed individuals and arcane exposed individuals decreased by 89.4% and 81.1% compared with that in stage 1, and the isolation rate of exposed persons in the hospital increased from 0 to 0.111, which was converted into 9 days. At the same time, the incidence rate decreased by 92.5% compared with that in stage 1, and the hospital admission time was 4.1 days faster than that in stage 1. It shows that with the improvement of medical capacity, isolation and tracking strategies, people’s self-protection awareness, and further understanding of COVID-19, the epidemic has been effectively controlled.

#### 3.1.3. Epidemic Fitting Results Analysis of Stage 3

On 18 February 2020, the existing confirmed cases reached a peak. With the arrival of the inflection point of the epidemic, the COVID-19 epidemic gradually declined. The simulation results of stage 3 are shown in Figure 8.

It can be seen from Figure 8 that the SEAIHR model and SEIR model fit the real data more accurately. It is because with the increase of prevention and control intensity and the improvement of medical detection means, the SEAIHR model can be approximated by the traditional SEIR model.

The evaluation results of the three models in stage 3 are shown in Table 7.

Table 7 shows that the SEAIHR model still has some improvement compared with classical models. The RMASE is 22.6% and 29.8% higher than the SIR and SEIR models, respectively, and the MAPE is 22.8% and 13.3% higher than the SIR and SEIR models, respectively.

When the best fitting result of stage 3 is obtained, the parameter values of the SEAIHR model are set as shown in Table 8.

Table 8 shows that the infection rate and incidence rate of arcane exposed persons are close to zero, and the incidence rate of hospitalization has increased by 38.9%. On the one hand, it shows that the tracking ability of exposed persons has been further enhanced, and on the other hand, the medical ability and the intensity of prevention and control can completely inhibit the further development of the COVID-19 epidemic.

In this paper, the experiment adopts the staged fitting method. The simulation results show that the SEAIHR model significantly improves the simulation accuracy compared with the classical model, and accurately reappears the trend of the COVID-19 epidemic outbreak to recession and the time node when the epidemic inflection point appears. However, since the arcane exposed population cannot be identified in advance, the isolation rate of such a population plays an important role in the accuracy of the SEAIHR model. In addition, the in-sample fit of the SEAIHR model outperforms the standard SIR/SEIR models, but it is not wholly surprising because the SEAIHR model has additional parameters and flexibility with respect to the competitor. In the actual selection of the models, we should use information criteria (such as AIC, BIC, or SIC) to select the optimal model by judging the fitting degree.

### 3.2. COVID-19 Spreading Trend Prediction

In Section 3.1, the SEAIHR model is used to trace the COVID-19 epidemic in Wuhan and verify the accuracy of the model in fitting in with historical data while revealing the COVID-19 epidemic development mechanism. In this section, in order to further verify the accuracy of the SEAIHR model in predicting the COVID-19 epidemic development under the normalization of prevention and control, the COVID-19 epidemic data published on the official website of the Xinjiang Health Commission from 16 July to 27 August 2020 was used for the experiment, which was divided into two parts: model parameter determination and COVID-19 epidemic development prediction.

#### 3.2.1. Data Fitting and Parameter Determination

In this paper, the cure rate was regarded as a time-varying parameter, and the COVID-19 epidemic data of Xinjiang, China, from 16 July to 27 August 2020 were used to fit and substitute the SEAIHR model. The fitting results of the cure rate are shown in Figure 9.

The simulation results comparison of different models with actual data is shown in Figure 10.

Figure 10 shows that the SEAIHR model is more accurate in fitting the actual data, and the fitting error is only 3.1%. Compared with the SIR and SEIR models, it accurately simulated the COVID-19 epidemic development and the change in the number of existing cases.

When the best fitting result is obtained, the parameter values of the SEAIHR model are set, as shown in Table 9.

As shown in Table 9, the value of $\lambda_{2}$ only accounts for 12.7% of $\lambda_{1}$, indicating that the government’s intervention measures played an important role in greatly reducing the infection risk of the exposed population. However, due to the existence of the incubation period of the virus, there is a time lag effect in the flow tracing of the suspect population, which would lead to the free spread of the virus in the population for a period of time before the onset of the disease, and the number of existing confirmed cases shows an upward trend in the initial two weeks.

#### 3.2.2. Epidemic Prediction

The model parameters were determined by using the actual data of the COVID-19 epidemic in Xinjiang and then were combined with the actual prevention and control measures to predict the future development of the COVID-19 epidemic. The experimental prediction result for the next four weeks is shown in Figure 11.

It can be seen from Figure 11 that the SEAIHR model is more accurate than the SIER model in predicting the time of the inflection point and the trend of epidemic decline. The inflection point of the epidemic occurred around 6 August, and the number of confirmed cases reached a peak at this time (a peak of about 771). It shows that the government’s various prevention and control measures have controlled the outbreak of the local epidemic in a relatively short period of time. Compared with the epidemic in Wuhan, Xinjiang only took about 20 days to control the epidemic, and the number of persons admitted to hospitals for confirmation did not exceed 800. It also can be seen that the number of confirmed cases was reduced to 150 at the end of August, indicating that the epidemic situation was basically stable.

Table 10 gives the model prediction results and the official actual data. It can be seen that the predicted peak occurrence date and values by the experiment are relatively consistent with the official statistical data.

In terms of the RMSE and MAPE, the SEAIHR model performs best on average. Considering the fact that the parameters of the SEAIHR model are time-varying, the mean square forecast error (MSFE) is used to estimate the accuracy of out-of-sample predictions. Starting from 16 July, the first-week COVID-19 epidemic data of Xinjiang is used to estimate the three models, and the epidemic development of the next 12 days is forecasted. Then a time series of the accuracy of the 12-day horizon forecasts for the three models is produced. A comparison of the MSFE of the three models is given in Table 11.

The proposed SEAIHR model and the estimation of its parameters are based on Chinese data and China’s epidemic prevention and control policy. However, the impact of different prevention and control measures on the spread of the COVID-19 epidemic will be completely different. Brauner et al. [20] gathered chronological data on the implementation of the COVID-19 pandemic with nonpharmaceutical interventions for several European countries and estimated the effectiveness of different nonpharmaceutical interventions. Davies et al. [21] examined the impact of tiered restrictions and alternatives for lockdown stringency, timing, and duration on SARS-CoV-2 transmission and hospital admissions and deaths from COVID-19 in England. Guaitoli et al. [22] studied whether Italy’s regional policies have effectively tackled the local infection risk arising from such heterogeneity. The impact of the epidemic prevention and control policies of these different countries on the spread of the COVID-19 pandemic is of great significance for further improving the proposed SEAIHR model.

## 4. Conclusions

On the basis of the analysis of COVID-19 virus spread characteristics, in this paper, an epidemic propagation model containing eight types of states is proposed. This paper analyzes and studies the epidemic situation from different development stages, and makes the model more consistent with the current epidemic situation through the change of control parameters. Therefore, the proposed model has a stronger generalization ability in different scenarios. The fitting and prediction experiments were carried out using the actual epidemic data of Wuhan and Xinjiang. The experimental results show that the fitting and prediction accuracy of the SEAIHR model is significantly improved compared to the classical models. However, the SEAIHR model is a time-varying model, it has additional parameters and flexibility with respect to the SIR and SEIR models. In the future, we will focus on measures of fits that penalize the number of parameters. On the other hand, the accuracy of the out-of-sample prediction of the SEAIHR model will be another future research focus.

## Figures and Tables

**Figure 1 ijerph-20-00132-f001:**
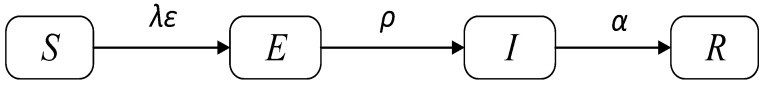
The transition between states of SEIR model.

**Figure 2 ijerph-20-00132-f002:**
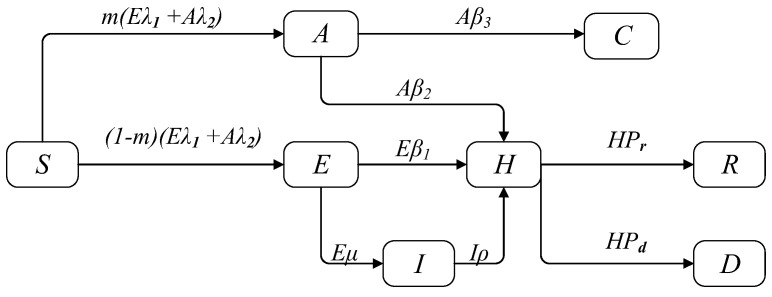
The transition between states of the SEAIHR model.

**Figure 3 ijerph-20-00132-f003:**
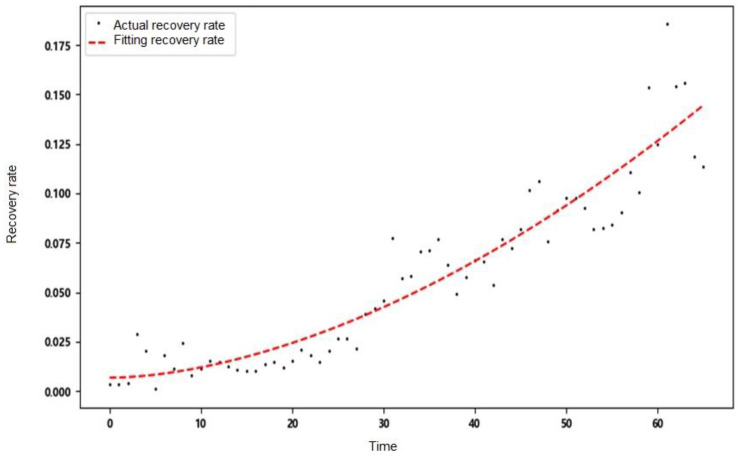
Fitting results of daily recovery rate.

**Figure 4 ijerph-20-00132-f004:**
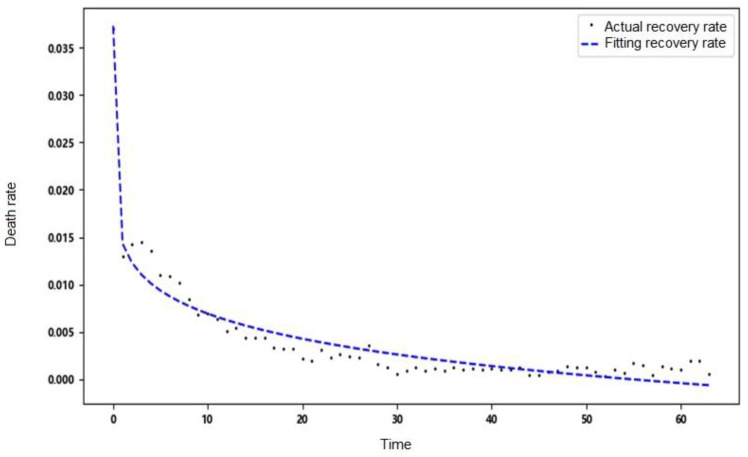
Fitting results of daily death rate.

**Figure 5 ijerph-20-00132-f005:**
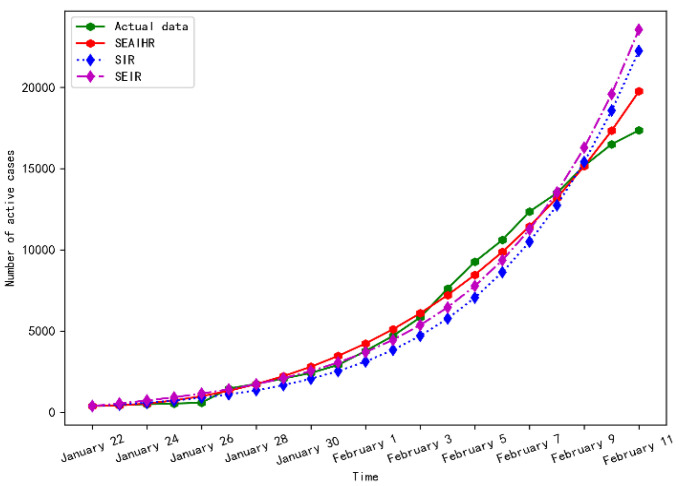
Simulation results comparison of stage 1.

**Figure 6 ijerph-20-00132-f006:**
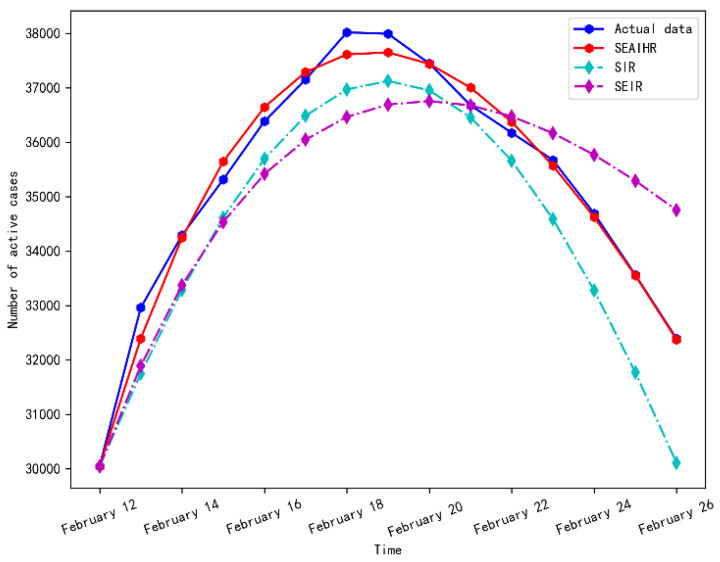
Simulation results comparison of stage 2.

**Figure 7 ijerph-20-00132-f007:**
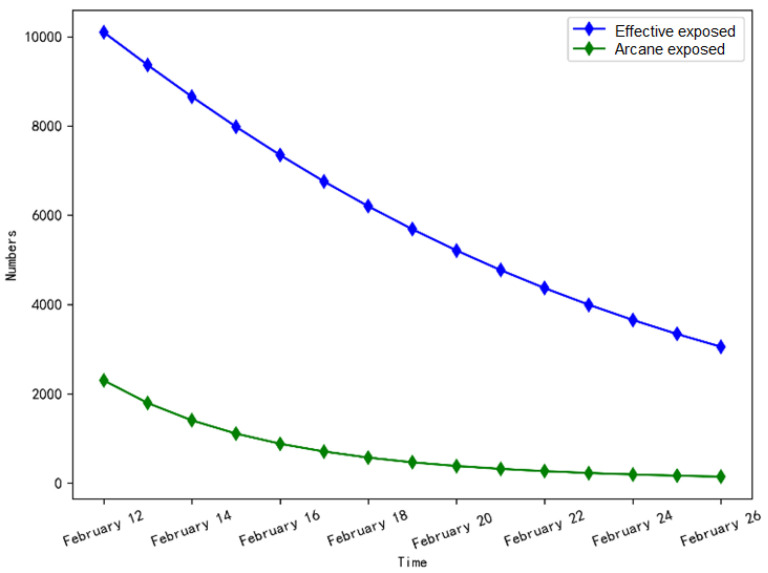
The number of exposed persons change in stage 2.

**Figure 8 ijerph-20-00132-f008:**
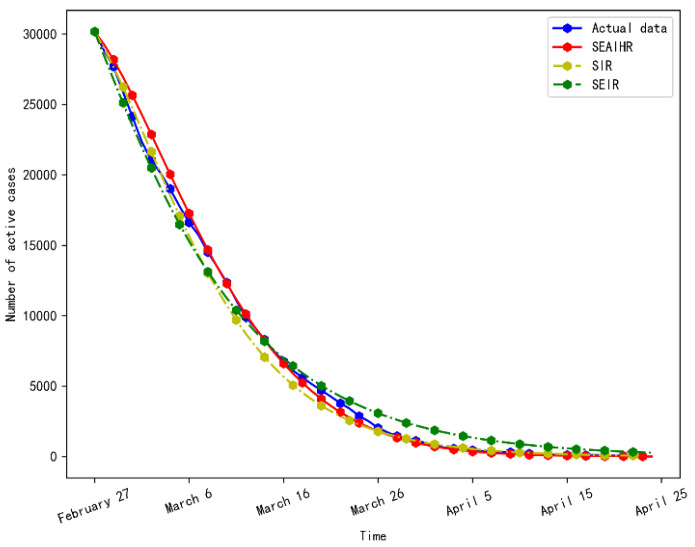
Simulation results comparison of stage 3.

**Figure 9 ijerph-20-00132-f009:**
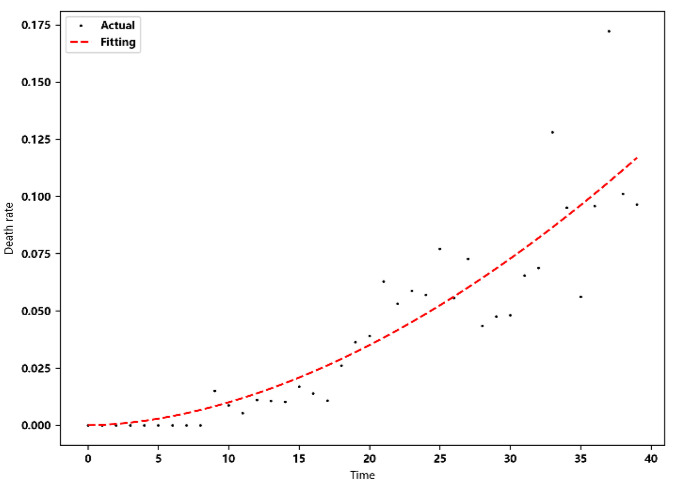
Fitting results of cure rate.

**Figure 10 ijerph-20-00132-f010:**
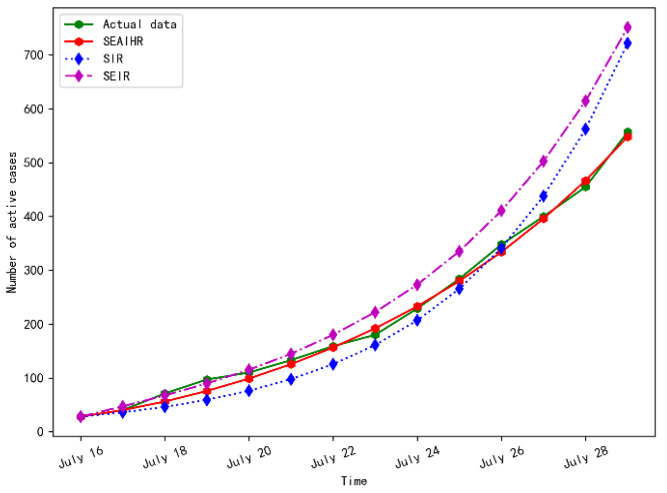
Simulation results comparison of different models.

**Figure 11 ijerph-20-00132-f011:**
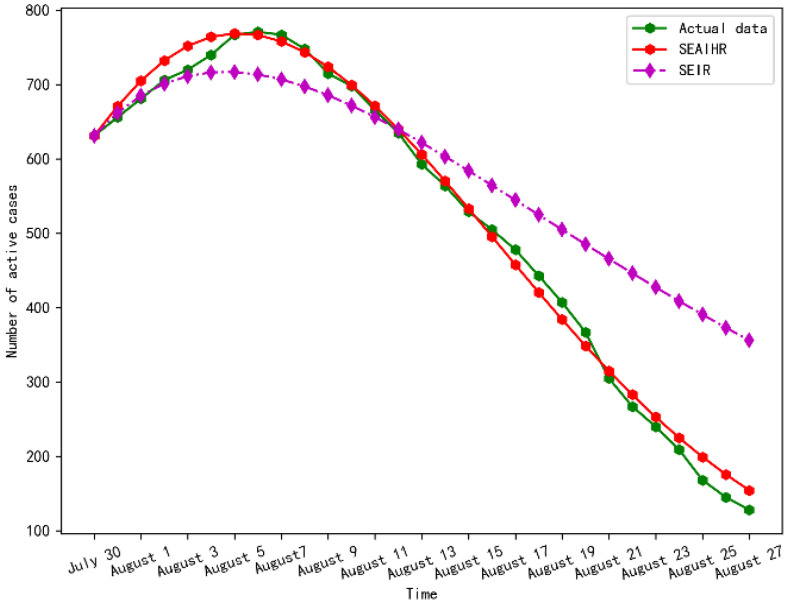
Epidemic prediction for the next four weeks.

**Table 1 ijerph-20-00132-t001:** Explanation of parameters.

Parameter Name	Explanation	Type
$\lambda_{1}$	The number of susceptible persons infected by an effective exposed person within a unit of time	Indirect parameter
$\lambda_{2}$	The number of susceptible persons infected by an arcane exposed person within a unit of time	Indirect parameter
$\beta_{1}$	The admission isolation rate of the effective exposed population	Direct parameter
$\beta_{2}$	The admission isolation rate of the arcane exposed population	Direct parameter
$\beta_{3}$	The daily recovery rate of the arcane exposed population	Direct parameter
μ	The probability of an effective exposed population transforming into an illness population in a unit of time	Direct parameter
ρ	Rate of isolation of illness population per unit time	Indirect parameter
$P_{r}$	The daily recovery rate of isolated inpatients	Time-varying parameter
$P_{d}$	The daily death rate of isolated inpatients	Time-varying parameter
m	The proportion of the arcane exposed population that lives among the exposed population	Direct parameter

**Table 2 ijerph-20-00132-t002:** Fitting results of recovery rate and death rate.

Fitting Equation	Fitting
*P_r_* = 0.0002271 × *t*^1.5413^	89.49%
*P_d_* = −0.021315 × *t*^0.148312^ + 0.037050	94.24%

**Table 3 ijerph-20-00132-t003:** Model evaluation in stage 1.

Model	RMSE	MAPE
SEAIHR	442	10.78%
SIR	1066	17.19%
SEIR	925	16.43%

**Table 4 ijerph-20-00132-t004:** Model estimated parameter values of stage 1.

Parameter	Value	Ratio of Standard Errors of Parameter
$\lambda_{1}$	0.491	4.32%
$\lambda_{2}$	0.433	5.16%
$\beta_{1}$	0	0.26%
$\beta_{2}$	0	0.28%
μ	0.4	0.58%
$\beta_{3}$	0.143	1.75%
ρ	0.09	5.39%
m	0.1	7.18%

**Table 5 ijerph-20-00132-t005:** Model evaluation in stage 2.

Model	RMSE	MAPE
SEAIHR	252	0.52%
SIR	521	1.08%
SEIR	960	2.24%

**Table 6 ijerph-20-00132-t006:** Model estimated parameter values of stage 2.

Parameter	Value	Ratio of Standard Errors of Parameter
$\lambda_{1}$	0.057	5.76%
$\lambda_{2}$	0.082	6.28%
$\beta_{1}$	0.111	0.33%
$\beta_{2}$	0.111	0.31%
μ	0.03	0.86%
$\beta_{3}$	0.143	1.65%
ρ	0.143	4.87%
m	0.1	6.88%

**Table 7 ijerph-20-00132-t007:** Model evaluation in stage 3.

Model	RMSE	MAPE
SEAIHR	548	12.52%
SIR	708	18.63%
SEIR	781	14.43%

**Table 8 ijerph-20-00132-t008:** Model estimated parameter values of stage 3.

Parameter	Value	Ratio of Standard Errors of Parameter
$\lambda_{1}$	0.021	0.66%
$\lambda_{2}$	0.002	2.37%
$\beta_{1}$	0.143	0.12%
$\beta_{2}$	0.143	0.12%
μ	0.001	0.18%
$\beta_{3}$	0.143	2.21%
ρ	0.23	2.69%
m	0.1	4.49%

**Table 9 ijerph-20-00132-t009:** Model estimated parameter values.

Parameter	Value	Ratio of Standard Errors of the Parameter
$\lambda_{1}$	1.34	6.28%
$\lambda_{2}$	0.171	3.36%
$\beta_{1}$	0.3	0.29%
$\beta_{2}$	0.3	0.27%
μ	0.45	0.66%
$\beta_{3}$	0.143	2.19%
ρ	0.33	3.75%
m	0.4	3.32%

**Table 10 ijerph-20-00132-t010:** Peak value comparison.

Peak Date	Data Source	Result
6 August	Actual data	771
5 August	SEAIHR simulation data	769
5 August	SEIR simulation data	715

**Table 11 ijerph-20-00132-t011:** Comparison of MSFE of three models.

Model	23 July–3 August	4–15 August	16–27 August
SEAIHR	16	9	4
SIR	64	144	225
SEIR	25	49	81

## Data Availability

Publicly available datasets were analyzed in this study. This data can be found here: http://wjw.wuhan.gov.cn (accessed on 26 April 2020).

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
