# Peer review of "Construction and Simulation Analysis of Epidemic Propagation Model Based on COVID-19 Characteristics"

_ijerph, 2022, doi:10.3390/ijerph20010132_

Round 1
Reviewer 1 Report
Based on the COVID-19 virus spread characteristics analysis, the author proposes an epidemic propagation model containing eight types of states. The paper analyses and studies the epidemic situation from different development stages and makes the model more consistent with the current epidemic by changing control parameters. Therefore, the proposed model has a more vital generalization ability in different scenarios. The fitting and prediction experiments were conducted using the epidemic data of Wuhan and Xinjiang. The experimental results show that the fitting and prediction accuracy of the proposed model is significantly improved compared with the SEIR model.
The paper can be considered for publication after some revision.
My comments are:
(1) The English of the paper should be improved.
(2) All equations should be punctuated.
(3) The author should check equation 5 and present it well.
(4) The figure 608328548 in the introduction should be written well.
(5) The motivation for the paper should be improved.
(6) The reference should be improved by https://doi.org/10.1016/j.chaos.2021.110885
(7) Why did the author not consider sensitivity analysis of critical parameters in the model?
(8) Can the author test their model with different country data, like the USA or India, to see the robustness of the proposed model?
Author Response
Dear Editor,
Thank you very much for your letter and for the comments by the reviewers. These comments are very valuable and helpful for my paper.
I appreciate the careful, constructive, and generally favorable reviews given to our paper by the reviewers.
I believe that I have adequately addressed all the excellent advices and questions raised by reviewers. Furthermore, I checked the manuscript and made sure the submitted manuscript is correct.
Response to the comments of reviewer 1:
Point 1: The English of the paper should be improved.
Response 1: Thanks for the comment. I had improved the English of the paper carefully.
Point 2: All equations should be punctuated.
Response 2: Thanks for the comment. I had added punctuation in all equations.
Point 3: The author should check equation 5 and present it well.
Response 3: Thanks for the comment. I had revised equation 5.
Point 4: The figure 608328548 in the introduction should be written well.
Response 4: Thanks for the comment. I had improved the figure 608328548 in introduction section carefully.
Point 5: The motivation for the paper should be improved.
Response 5: Thanks for the comment. I had improved the motivation for the paper in introduction section carefully.
Point 6: The reference should be improved by https://doi.org/10.1016/j.chaos.2021.’
Response 6: Thanks for the comment. I had improved the reference.
Point 7: Why did the author not consider sensitivity analysis of critical parameters in the model.
Response 7: Thanks for the comment. The concept of sensitivity analysis is used to obtain the relative importance of each model parameter. Because the important parameters in this model are obtained through the data analysis and fitting of historical data, so I didn’t consider sensitivity analysis of critical parameters in the model. I had read carefully the paper “Sensitivity assessment and optimal economic evaluation of a new COVID-19 compartmental epidemic model with control interventions”, which gave me a good idea for future research. The sensitivity analysis of critical parameters in the proposed model would become an important research direction in my future.
Point 8: Can the author test their model with different country data, like the USA or India, to see the robustness of the proposed model?
Response 8: Thanks for the comment. Because the epidemic prevention and control measures are different in each country and region, this paper only uses the epidemic data of China for testing. In the future, more countries' epidemic data will be collected to test the robustness of the model.

Reviewer 2 Report
See attached file.

Author Response
Dear Editor,
Thank you very much for your letter and for the comments by the reviewers. These comments are very valuable and helpful for my paper.
I appreciate the careful, constructive, and generally favorable reviews given to our paper by the reviewers.
I believe that I have adequately addressed all the excellent advices and questions raised by reviewers. Furthermore, I checked the manuscript and made sure the submitted manuscript is correct.
Response to the comments of reviewer 1:
Point 1: It is suggested that the author comment more in detail on the consistency of the proposed models, the relations in-between them and also that a detailed revision of notation and some formulas be performed.
The model of eqn. (1), which is the basis for the later study, has several points to be explained. So, if the infection is not totally removed, all the population becomes recovered asymptotically as time tends to infinity which seems to be contradictory. It has to be also pointed out that, even if I(t) vanishes asymptotically and, depending on the profile the function vanishes, R(t) could also diverges as time tends to infinity since the vanishing condition to avoid that possibility is necessary but not sufficient.
Response 1: Thanks for the comment. SEIR model is a classical propagation model of infectious disease. The propagation dynamics equation of the model is shown in equation (1). The total number of people in the model is N, regardless of birth and death, immigration and emigration, and the total number remains unchanged. At t time, S(t) + E(t) + I(t) + R(t)=1. So the situation that the all the population is jointly recovered and susceptible as time tends to infinity while, at the same time, the infection is not removed will not happen.
Point 2: Why are the populations of the model (2) subscripted with“t" in the time-derivatives but not in the right-hand- sides of the differential equations?. Are not the same populations of the right-hand-sides those who timed derivatives influence the differential equations to build the trajectory solutions?. However, in the model (1), they are subscripted in both sides of the equations.
Response 2: Thanks for the comment. I am sorry that the differential equations of model (2) is wrong. I had revised the differential equations.
Point 3: Which is the relation of model (2) to model (1)? . Please, explain it more in detail.
Response 3: Thanks for the comment. Model (2) is constructed by adding three new states on the basis of model (1). I had added detailed explanation in paper.
Point 4: Table 1: the left-hand-sides are proportions. How, is it interpreted that P/d is negative after a certain time Note that P/d (t) is continuous and that the negative term is strictly decreasing with time while the positive one is constant.
Response 4: Thanks for the comment..represents daily death rate of isolated inpatients, the probability will only change over time, it will not be a negative number.
Point 5: The least-squares algorithm is runned for discrete data- However, the epidemic model is of a continuous-time nature . How is the model adapted for estimation purposes involving discrete data?
Response 5: Thanks for the comment. In the experiment, I take each date as a discrete point to get the number of newly recovery people and newly death toll on that day, so as to use nonlinear least square method for fitting.
Point 6: Line 147:“square"-"squares".
Response 6: Thanks for the comment. I had revised the mistake.
Point 7: Eqn.5: It seems that there is a parenthesis missed to calculate the square in the right-hand-side.
Response 7: Thanks for the comment. I had revised Eqn. 5.
Point 8: Eqns. (5)-(6) : there are right-hand-side summations under a summation index “i” which is not reflected in the right-hand-sides.
Response 8: Thanks for the comment. I had revised Eqn. 5 and Eqn. 6.
Point 9: The conclusions section should be extended with more technical details to comment on the details of the proposed approach.
Response 9: Thanks for the comment. I had improved conclusion section.
Point 10: There is some previous background literature, which follows, which also considers the dead population as an extra subpopulation in epidemic models which should be incorporated to the list of references and briefly commented in the manuscript.
Response 10: Thanks for the comment. I had added related literatures.

Reviewer 3 Report
In this paper, the author introduces a variation of the standard SIR/SEIR models (and their variations) by augmenting a few compartments that include realistic assumptions on illness and hospitalization (SEAIHR model). In addition, the author assumes that the underlying parameters of the model are time-varying. The paper's main message is to show that the proposed SEAIHR model better fits the data than the conventional SIR and SEIR models when estimated to the different phases of the covid epidemic in China.
I enjoyed reading the paper and have a few comments related to the model, empirical exercise and its relevance.
(1) The model and how it is brought to the data
As already mentioned, the author proposes a contagion model augmented with a few additional features, namely the existence of Arcane exposed population and Illness and Hospitalization status. These assumptions make sense, so the theoretical background behind the proposed model is sound.
However, I need some clarification about identifying the parameters that determine the flows between the newly proposed status from the data. How the data on hospitalization can distinguish between arcane (non-symptomatic) and exposed (symptomatic) population needs to be clarified. Which type of data is available to separately identified the two flows? My doubt is confirmed by the estimated beta_1 and beta_2 displayed in the tables: those parameters are always identical. I am confident that this fact does not happen by chance.
If these two parameters are bounded to be identical because it is impossible to distinguish them with the available data, then the author should clarify this issue upfront and discuss its theoretical consequences and interpretation.
In addition, the author should provide standard errors for the parameter estimates. I am still not convinced that all the parameters indirectly estimated are well identified. But, of course, I might be wrong: computing and displaying the standard errors will reassure me and the audience.
2) Fit
The main body of the paper shows that the in-sample fit of the SEAIHR model outperforms one of the standard SIR/SEIR models. The author shows that the gain fit is quite sizeable, but that is not fully surprising because the proposed model has additional parameters and flexibility with respect to the competitor. In other words, the RMSE of the SEAIHR model cannot be lower than the nested SIR/SEAIR. For this reason, the author should focus on measures of fits that penalize for the number of parameters: information criteria (BIC or SIC, for example) address this problem.
The good news is that because of the magnitude of the fit gain, those information criteria should select the SEAIHR model as the best-penalized fit, but this fact must be adequately and formally established.
3) Prediction
I am most concerned about using the SEAIHR model with time-varying parameters for making predictions. As the author points out, an epidemic model is valuable if it can help predict future epidemic patterns. However, the fact that parameters are time-varying undermines this property unless those parameters have a predictable law of motion, which is not specified and left unconstrained in this paper.
The authors should be upfront and discuss this caveat. More generally, while pointing in the right direction, I think the exercise in section 3.2.2 needs to be more formal about the out-of-sample prediction properties of the model because it focuses on a single episode with no proper out-of-sample forecast evaluation.
Here is what the author should do to solve this issue:
a- Consider the three models SIR/SEIR/SEAIHR
b- Select a time horizon starting from February 12, for example, the first week week (February 12- February 19)
c- Estimate the three models using only those data
d- Forecast out of sample the path of the forecasted variables up to an h-horizon. For example, if h=14, compute the out-of-sample forecast for the next 14 days for each model.
e- Compute a proper measure of goodness of the forecast using only the out-of-sample prediction up to horizon h, for example, the mean square forecasted error.
f- Augment the in-sample window using additional data in the estimation, for example, adding one week, thus considering Feb12-Feb26; reestimate the three models; recompute the MSFE for the h-horizon out-of-sample prediction, and keep on doing that for all the data available
This process will produce a time series of the accuracy of the 14-day horizon forecasts for the three models in the whole epidemic wave.
If the SEAIHR model has better out-of-sample performance, it will be clearly displayed in the comparison.
To conclude, some improvements in the execution of the paper is needed, in my opinion.
Author Response
Dear Editor,
Thank you very much for your letter and for the comments by the reviewers. These comments are very valuable and helpful for my paper.
I appreciate the careful, constructive, and generally favorable reviews given to our paper by the reviewers.
I believe that I have adequately addressed all the excellent advices and questions raised by reviewers. Furthermore, I checked the manuscript and made sure the submitted manuscript is correct.
Response to the comments of reviewer 3:
Point 1: The model and how it is brought to the data
As already mentioned, the author proposes a contagion model augmented with a few additional features, namely the existence of Arcane exposed population and Illness and Hospitalization status. These assumptions make sense, so the theoretical background behind the proposed model is sound.
However, I need some clarification about identifying the parameters that determine the flows between the newly proposed status from the data. How the data on hospitalization can distinguish between arcane (non-symptomatic) and exposed (symptomatic) population needs to be clarified. Which type of data is available to separately identified the two flows? My doubt is confirmed by the estimated beta_1 and beta_2 displayed in the tables: those parameters are always identical. I am confident that this fact does not happen by chance.
If these two parameters are bounded to be identical because it is impossible to distinguish them with the available data, then the author should clarify this issue upfront and discuss its theoretical consequences and interpretation.
In addition, the author should provide standard errors for the parameter estimates. I am still not convinced that all the parameters indirectly estimated are well identified. But, of course, I might be wrong: computing and displaying the standard errors will reassure me and the audience.
Response 1: Thanks for the comment. The arcane (non-symptomatic) and exposed (symptomatic) population can be distinguished only after being confirmed. Parameter and indicates the admission isolation rate of these two groups. Therefore, these two parameters can only be set to the same value during simulation. For this point, an explanation and discussion has been added in the end of Section 3.1. In order to obtain estimates of the standard errors of parameters in SEAIHR model, 500 simulations of SEAIHR model are obtained. And ratio of standard errors of parameter is given in Table.
Point 2: 2) Fit
The main body of the paper shows that the in-sample fit of the SEAIHR model outperforms one of the standard SIR/SEIR models. The author shows that the gain fit is quite sizeable, but that is not fully surprising because the proposed model has additional parameters and flexibility with respect to the competitor. In other words, the RMSE of the SEAIHR model cannot be lower than the nested SIR/SEAIR. For this reason, the author should focus on measures of fits that penalize for the number of parameters: information criteria (BIC or SIC, for example) address this problem.
The good news is that because of the magnitude of the fit gain, those information criteria should select the SEAIHR model as the best-penalized fit, but this fact must be adequately and formally established.
Response 2: Thanks for the comment. The statistical idea followed by statistics such as Bayesian Information Criterion is to impose "penalty" according to the number of independent variables while considering fitting residuals. For SEAIHR model, because of the addition of additional parameters, the fitting effect is much better than other traditional models. It is also the reason for the newly added states in the proposed SEAIHR model in this paper. Your suggestion is very valuable. I had added the description and explanation of this fact in the end of Section 3.1.
Point 3: 3) Prediction
I am most concerned about using the SEAIHR model with time-varying parameters for making predictions. As the author points out, an epidemic model is valuable if it can help predict future epidemic patterns. However, the fact that parameters are time-varying undermines this property unless those parameters have a predictable law of motion, which is not specified and left unconstrained in this paper.
The authors should be upfront and discuss this caveat. More generally, while pointing in the right direction, I think the exercise in section 3.2.2 needs to be more formal about the out-of-sample prediction properties of the model because it focuses on a single episode with no proper out-of-sample forecast evaluation.
Here is what the author should do to solve this issue:
a- Consider the three models SIR/SEIR/SEAIHR
b- Select a time horizon starting from February 12, for example, the first week week (February 12- February 19)
c- Estimate the three models using only those data
d- Forecast out of sample the path of the forecasted variables up to an h-horizon. For example, if h=14, compute the out-of-sample forecast for the next 14 days for each model.
e- Compute a proper measure of goodness of the forecast using only the out-of-sample prediction up to horizon h, for example, the mean square forecasted error.
f- Augment the in-sample window using additional data in the estimation, for example, adding one week, thus considering Feb12-Feb26; reestimate the three models; recompute the MSFE for the h-horizon out-of-sample prediction, and keep on doing that for all the data available
This process will produce a time series of the accuracy of the 14-day horizon forecasts for the three models in the whole epidemic wave.
If the SEAIHR model has better out-of-sample performance, it will be clearly displayed in the comparison.
Response 3: Thanks for the comment. Relevant experiments had added according to the steps you have given. Mean square forecasted error ((MSFE) is used to estimate the accuracy of out-of-sample prediction of the three models, and Table 11gives the comparison results.

Round 2
Reviewer 1 Report
Congratulations
Author Response
Thank you for your comments.
Reviewer 2 Report
The paper has been improved and it can be accepted.
Author Response
Thank you for your comments.
Reviewer 3 Report
Dear author, Thanks for sending the revision. I have found the revised version of the paper very much improved. The only remark I would add, in the discussion/conclusion section, is that the proposed SEAIHR model and the estimation of its parameters using Chinese data, do not include the possible (and likely) effects that non-pharmaceutical interventions might have (and have had), even at the sub-regional level, in the propagation of the virus. There are several papers that have identified these effects, which, in principle could be important (see below a list of three papers that make this point). I am totally fine for the author to ignore these effects in their model, but two lines in the paper, as a possible limitation of the study, could be valuable. Besides this tiny change, I believe the paper is now suitable for publication. - J.M. Brauner, S. Mindermann, M. Sharma, D. Johnston, J. Salvatier, T.Gavenčiak, et al. Inferring the effectiveness of government interventions against covid-19 Science (2020) - N.G. Davies, R.C. Barnard, C.I. Jarvis, T.W. Russell, M.G. Semple, M. Jit, et al. Association of tiered restrictions and a second lockdown with covid-19 deaths and hospital admissions in England: a modelling study The Lancet Infectious Diseases (2020) - G.Guaitoli, R. Pancrazi.Covid-19: Regional policies and local infection risk: Evidence from Italy with a modelling study.
The Lancet Regional Health - Europe (2021)
Author Response
Thank you for your comments. These three articles are very valuable for further improving the proposed model in this paper. At the end of the experimental section of this paper, the analysis and quotation of these three articles is added, pointing out the limitations of the SEAIHR model and that different prevention and control measures will have completely different effects on the spread of the COVID-19.